# GABAergic neurons are susceptible to BAX-dependent apoptosis following isoflurane exposure in the neonatal period

**Andrew M. Slupe**[1], **Laura Villasana**[1], **Kevin M. Wright**[2]*

**1** Department of Anesthesiology and Perioperative Medicine, Oregon Health & Science University, Portland, Oregon, United States of America, **2** Vollum Institute, Oregon Health & Science University, Portland, Oregon, United States of America

* wrighke@ohsu.edu

**Data Availability Statement:** All relevant data are within the manuscript and its Supporting information files.

**Funding:** This work was supported by Foundation for Anesthesia Education and Research (FAER)

## Abstract

Exposure to volatile anesthetics during the neonatal period results in acute neuron death. Prior work suggests that apoptosis is the dominant mechanism mediating neuron death. We show that *Bax* deficiency blocks neuronal death following exposure to isoflurane during the neonatal period. Blocking Bax-mediated neuron death attenuated the neuroinflammatory response of microglia following isoflurane exposure. We find that GABAergic interneurons are disproportionately overrepresented among dying neurons. Despite the increase in neuronal apoptosis induced by isoflurane exposure during the neonatal period, seizure susceptibility, spatial memory retention, and contextual fear memory were unaffected later in life. However, *Bax* deficiency alone led to mild deficiencies in spatial memory and contextual fear memory, suggesting that normal developmental apoptotic death is important for cognitive function. Collectively, these findings show that while GABAergic neurons in the neonatal brain undergo elevated Bax-dependent apoptotic cell death following exposure to isoflurane, this does not appear to have long-lasting consequences on overall neurological function later in life.

## Introduction

Exposure to volatile anesthetics in early life has consistently been found to result in widespread neuron death in small mammal and non-human primate models [1]. Vulnerability to anesthesia-associated neuronal death is confined to a temporal window coincident with high levels of brain growth and synaptogenesis [2]. Human correlates of these processes suggest that vulnerability to injury for pediatric patients receiving anesthesia may extend into early childhood [3, 4]. Neuronal injury by volatile anesthetics may be responsible for lasting behavioral and learning deficits in children exposed to the perioperative environment [5]. However, the molecular pathways involved in neuron death following exposure to volatile anesthetics in early life remain unclear. Further, the relative contribution to injury of disrupted cellular processes in surviving neurons following exposure to volatile anesthetics is unknown.

Several prior observations have suggested that apoptosis may be the dominant mechanism of death of neurons exposed to volatile anesthetics in the neonatal period. Anesthesia exposure

Research Fellowship Grant (RFG-02-15-17) to
AMS and National Institutes of Health/National
Institute of Neurological Disorders and Stroke
(NIH/NINDS) Grant (R01 NS091027) to KMW.
https://www.asahq.org/faer https://www.ninds.nih.
gov/ The funders had no role in study design, data
collection and analysis, decision to publish, or
preparation of the manuscript.

**Competing interests:** The authors have declared
that no competing interests exist.

alters the expression ratio of Bcl-2 family members in favor of apoptosis through down-regulation of anti-apoptotic Bcl-2 and Bcl-x(L) and up regulation of pro-apoptotic Bax and Bad [2, 6–8]. The morphology of dying neurons is consistent with apoptotic cell death [9]. Finally, early life exposure to ethanol, which has a mechanism of action believed to be similar to volatile anesthetics, results in neuroapoptosis [10]. These studies suggest, but do not definitively demonstrate, that neuron death following exposure to volatile anesthetics in early life occurs by apoptosis.

The proapoptotic Bcl-2 family member Bax is activated by signaling events in cells undergoing apoptotic cell death [11]. In neurons, genetic deletion of *Bax* is sufficient to block activation of the apoptotic pathway in response to multiple stimuli, including trophic factor withdrawal, excitotoxic insult, and ethanol exposure [10, 12, 13]. Blocking Bax is therefore a putative target to delineate neuron death occurring through activation of the apoptosis pathway from other mechanisms of injury.

Vulnerability to anesthesia-associated death may not be uniform across neuronal types. It has been suggested that GABAergic interneurons in superficial cortical layers may be overrepresented among dying neurons [9]. This population undergoes significant cellular pruning by apoptosis during the same developmental window as vulnerability to anesthesia [14]. It is unknown how anesthesia impacts this normal physiological cell loss and what the consequences of neuron loss may be on neuronal circuit function.

Previous work has shown that early life exposure to ethanol leads to Bax-mediated neuronal apoptosis, followed by a neuroinflammatory response characterized by microglial activation [15]. A similar inflammatory profile following exposure to volatile anesthetics has been observed [16]. However, it is unknown if the neuroinflammatory response to volatile anesthetics occurs as a consequence of neuron death, or if it is an independent process that contributes to neuron death.

To investigate the mechanism of neurotoxicity following exposure to isoflurane in early life we assessed death and neuroinflammation in neonatal *Bax* knockout mice. We profiled GABAergic neuron vulnerability to anesthesia-induced death, and probed for a disruption of global inhibitory-excitatory balance later in life. Finally, we used behavioral assays to determine the consequences of neuronal loss due to neonatal exposure to anesthesia on cognitive function later in life.

## Materials and methods

### Animals and anesthetic exposure

All procedures involving animals were approved by the Oregon Health and Science University Institutional Animal Care and Use Committee. Animals were allowed ad lib access to food and water and maintained in standard 12-hour light-dark cycle. Heterozygous *Bax* mice (B6.129X1-Bax$^{tm1Sjk}$lj, stock #002994), *Bax$^{Flox}$*;*Bak$^{-/-}$* (B6;129-Baxtm2Sjk Bak1tm1Thsn/J, stock #006329), *R26LSL-TdTomato* (B6.Cg-Gt(ROSA)26Sortm9(CAG-tdTomato)Hze/J, stock #007909), and *Gad2-IRES-Cre* (Gad2tm2(cre)Zjh/J, stock #010802) were obtained from The Jackson Laboratory. A conditional GABAergic interneuron specific *Bax* knock-out/reporter line, *Gad2-IRES$^{Cre}$*;*Bax$^{Flox;TdTom/TdTom}$*, was generated by interbreeding of *Bax$^{Flox}$*;*Bak$^{-/-}$*, *R26LSL-TdTomato* and *Gad2-IRES-Cre*. The breeding strategy resulted in return of the wild-type *Bak* allele. The experimental population was generated by breeding *Gad2-IRES$^{Cre/Cre}$*;*Bax$^{Flox/Flox}$*;*R26$^{TdTom/TdTom}$* mice with *Bax$^{Flox/+;TdTom/TdTom}$*. Heterozygosity of the *Gad2* allele was maintained in the experimental population as altered seizure threshold has been reported in homozygous *Gad2-IRES-Cre* animals [17]. We found that on occasion, the *Gad2-IRES-Cre* line exhibited inappropriate recombination that resulted in a mosaic pattern throughout the

animal. Therefore, proper recombination by the *Gad2-IRES-Cre* line was verified by expression of tomato specifically in GABAergic populations in all experimental animals.

On PND 7, neonatal mice were exposed to isoflurane partially titrated to a level of 1 MAC for 6 hours. At the beginning of the 12-hour light cycle animals were placed on soft bedding in an acrylic induction chamber on circulating water bath heaters such that chamber temperature was maintained at 34°C. Humidity within the chamber was maintained with an open water bath placed near the common gas inlet. Carrier gas was composed of medical air and oxygen at an $FiO_2$ of 50% verified with a $MaxO_2$ ME oxygen sensor (MaxTec, Salt Lake City, UT) delivered at 1 LPM. Isoflurane concentration within the induction chamber was monitored using a POET II gas analyzer (Criticare Technologies, Inc, North Kingsworth, RI) and sampling from the chamber exhaust port. Our experience and that reported elsewhere suggests that potency of isoflurane increases with prolonged exposure in rodents with resultant high levels of mortality during exposure without titration [18]. For this reason the exposure paradigm used here was partially titrated to a level of 1 MAC guided by previous reports [19]. Specifically, a general anesthetic state was induced by exposure to 4% isoflurane for 15 minutes, this was reduced to 2% for an additional 15 minutes, then further reduced by 0.2% increments every 30 minutes to a final concentration of 1% for the remaining time. At 150 minutes of exposure, responsiveness of the exposed population was assessed by toe pinch and it was found that ~50% of the exposed animals responded with movement following stimulation. This exposure paradigm was associated with a 1% mortality rate. Control littermates were maintained in similar conditions without isoflurane, all responded to toe pinch at 150 minutes and mortality rate was 0%. Arterial blood gas analysis was not performed due to institutional prohibitions against un-anesthetized terminal blood sampling. Following completion of the 6 hour exposure isoflurane administration was discontinued, carrier gas flow continued and mice were allowed to recover for 20 minutes following complete washout of isoflurane from the induction chamber. At the end of 20 minutes, responsiveness to toe pinch with vocalization and/or purposeful movement was confirmed and animals were returned to their home cage.

## Genotype and sex determination

Tissue samples derived from toe tagging were used for PCR. DNA was extracted with Extracta DNA Prep kit (QuataBio, Beverly, MA) and PCR reactions were performed using genotype-specific primer sets. Sex was determined using primers directed towards the sex-associated gene *Ube1y1* as previously described [20].

## Neuroapoptosis assessment

Animals were euthanized by rapid decapitation two hours after conclusion of the isoflurane exposure. Brains were isolated and preserved in 4% PFA for 16–24 hours at 4°C. A Leica VT1200 Vibratome was used to obtain 50 μm thick coronal sections from Paxinos plate P6 #25 to #34 [21], every fifth section was collected for analysis. For immunohistochemistry free-floating sections were blocked and permeabilized with 2% donkey serum in 0.2% Triton X-100/PBS for four hours at room temperature followed by antibody staining with 1:1000 anti-cleaved caspase 3 (Cell Signaling Technology) and subsequently 1:1000 anti-rabbit Alexa 568 (ThermoFisher) and 1:5000 Hoechst 33342 (ThermoFisher). Sections were mounted on PermaFrost Plus slides (ThermoFisher), coated in FluoroMount-G (SouthernBiotech) and cover-slipped. For Fluorojade C staining sections were mounted to PermaFrost Plus slides and air dried for 24 hours. If not immediately processed, these slides were stored at -80°C. Fluorojade C staining was carried out as previously described [22]. Following staining sections were

coated in DPX Mounting Media (SigmaAldrich) and coverslipped. Sections were imaged with a Zeiss Axio Imager M2 upright microscope equipped with an ApoTome.2. Immunohistochemical analysis was performed using Fiji (NIH) by an observer blind to genotype and experimental conditions.

## Microglia activation assessment

Microglia morphology, Iba1 content and cytokine gene expression was determined following the initiation of the isoflurane exposure. For immunohistochemistry, brains were isolated and processed as described above 24 hours following the initiation of isoflurane exposure. Iba1 containing cells were labeled by antibody staining with 1:500 anti-Iba1 (Wako #019–19741) and subsequently 1:1000 anti-rabbit Alexa 568 and 1:5000 Hoechst 33342. Sections were imaged using a Zeiss Axio Imager M2 upright microscope equipped with an ApoTome.2 at low (10x) magnification. High magnification images were collected as a 0.5 μm z-stack with a Nikon A1R confocal microscope with an optical magnification of 60x and digital magnification of 3.5x and images were processed as a projection through the stack using Fiji.

Cortical Iba1 content was determined by western blot 24 hours following the initiation of isoflurane exposure. Cortical hemispheres including the hippocampus were surgically isolated and snap frozen in liquid nitrogen and stored at -80˚C. Tissue was lysed in 2 ml of 20 mM Tris HCl pH 7.5, 150 mM NaCl, 1 mM EDTA, 1% Triton X-100 and 1xHALT Protease and Phosphatase Inhibitor Cocktail (ThermoFisher) per hemisphere and processed with a dounce homogenizer. Insoluble material was removed by centrifugation (8000 xg for 15 minutes) and discarded. Protein concentration was determined with a Pierce BCA Protein Assay Kit (ThermoFisher) and 5 μg protein/sample were separated on 15% SDS-PAGE gels. Protein samples were transferred to PVDF membranes (ThermoFisher) and blocked in TBS Odyssey Blocking Buffer (Li-Cor). Membranes were probed with 1:2000 anti-Erk2 (Santa Cruz Biotechnology sc-1647), 1:1000 anti-Bax (CST D3R2M), and 1:500 anti-Iba1 (Wako #019–19741) followed by species-specific anti-IgG IR800CW or IR680CW secondary antibodies (Li-Cor). Erk was used as a loading control as it's expression has previously been demonstrated to be insensitive to ethanol exposure in the neonatal period [23] Samples were imaged using an Odyssey CLx system (Li-Cor). Band density quantification was performed using Fiji.

Markers of microglia activation and microglia-derived inflammatory cytokine expression were assessed by RT-PCR 12 hours after initiation of isoflurane exposure. Cortical hemispheres including the hippocampus were surgically isolated and RNA was isolated using an RNeasy Mini Kit (Qiagen). RNA concentration was determined using a NanoDrop (ThermoFisher), and 4 μg RNA was used for cDNA synthesis with a SuperScript III First-Strand Synthesis System using random hexamer primers (ThermoFisher). Template cDNA was diluted 1:5 in nuclease free $H_2O$ and stored at -20˚C until PCR which was performed within 7 days. Uniplex qPCR was performed using TaqMan Fast Advanced Master Mix and FAM labeled primers (ThermoFisher) with a ViiA 7 System (Applied BioSystems). The UBE2D2 gene transcript was used as the internal reference, as it has previously been demonstrated to be relatively resistant to expression alteration in neurotoxic states [24].

## GABAergic interneuron quantification and selective protection

PND 7 mice from $Gad2^{IRES-Cre/+}$;$Bax^{Flox}$; $R26^{TdTom/TdTom}$ line were used to determine the relative cortical GABAergic interneuron population size by FACS. Cortical hemispheres including the hippocampi were isolated in Hank's Buffered Salt Solution (HBSS) at 4˚C and the meninges were removed. Hemispheres were then minced to ~1 mm$^3$ pieces and digested in 1ml of HBSS, 10 units papain (Roche), 5 mM L-cysteine, and 50 units DNase (Promega) for 15

minutes at 37°C with agitation. Digestion was halted by the addition of 100 μl fetal bovine serum (ThermoFisher). Neurons were dissociated by trituration with three flame polished pasteur pipettes of decreasing internal diameter. The cell suspension was then filtered through a 40 μm cell strainer. The cell suspension was then floated on top of 5 ml 20% Percoll (Sigma) in HBSS and centrifuged (800 xg for 5 minutes). The supernatant was discarded, and the pellet resuspended in 0.5 ml HBSS. TdTomato positive and negative neuron populations were counted and sorted by RFP fluorescence, forward scatter and side scatter gates using a Becton Dickinson InFlux cell sorter (OHSU Flow Cytometry Core), 450K to 1000K cells were collected per replicate experiment. A sample of the FACS input cell suspension as well as the sorted cells were lysed with 100 μl/100K cells of the lysis buffer described above and subjected to 10% SDS-PAGE and western blot with 1:500 anti-RFP (Rockland 8E5.G7) and 1:1000 anti-Bax antibodies as described above.

To evaluate relative vulnerability of the GABAergic interneuron population versus the non-GABAergic interneuron population following exposure to isoflurane by immunohistochemistry, $Gad2^{IRES-Cre/+}$;$Bax^{Flox/Flox}$; $R26^{TdTom/TdTom}$ and $Gad2^{IRES-Cre/+}$;$Bax^{Flox/+}$; $R26^{TdTom/TdTom}$ animals were exposed to isoflurane or control conditions as described above. Two hours after the conclusion of the exposure, animals were euthanized and brains sections prepared as described above. Sections were stained with 1:500 anti-RFP and 1:1000 anti-cleaved caspase 3. Cleaved caspase 3 positive neurons in cortical layers II/III were counted and the proportion of GABAergic neurons (TdTomato positive) determined by an investigator blinded to the experimental conditions.

## Seizure susceptibility assay

To investigate whether neonatal exposure to anesthesia influenced seizure susceptibility, we used a flurothyl exposure paradigm as previously described [25, 26]. $Gad2^{IRES-Cre/+}$;$Bax^{Flox/Flox}$; $R26^{TdTom/TdTom}$ and $Gad2^{IRES-Cre/+}$;$Bax^{Flox/+}$; $R26^{TdTom/TdTom}$ animals were exposed to isoflurane or control conditions on PND7, then returned to their home cages and allowed to age until undergoing seizure susceptibility testing during postnatal week (PNW) 7–8 as previously described [25, 26]. Briefly, animals were placed in an enclosed cylindrical chamber 6 inches in diameter and 10 inches tall with a physically isolated vaporization chamber. Liquid 10% Bis (2,2,2-Trifluoroethyl) Ether in 85% EtOH/5% H2O solution was delivered to the vaporization chamber at a rate of 6 ml/hr. Time to the onset of the first myoclonic jerk and generalized tonic-clonic seizure (TCS) as evidenced by full body convulsant movements with loss of postural control was recorded by an observer blind to the experimental conditions. Immediately following the onset of TCS the animals were removed from the exposure chamber, resulting in spontaneous seizure cessation within 5 seconds. Animals recovered to their baseline activity status over 5 minutes and were then returned to their home cages.

## Behavioral assays

The long-term consequences on learning and memory following early life exposure to isoflurane were assessed. A separate cohort of animals from the seizure paradigm cohort were exposed to isoflurane on PND 7 and aged to PNW 10–11, at which time they underwent behavioral tests. Hippocampal-dependent visual-spatial learning and memory was assessed by Morris-Water-Maze (MWM) on PNW 10. Hippocampal-and amygdala-dependent contextual fear memory formation and amygdala-dependent cued fear memory were assessed on PNW 11.

MWM testing was performed generally as previously described [27]. Mice were placed in a large water bath (122cm wide; 20°C ± 1) surrounded by prominent visual cues and were removed upon locating the hidden platform (submerged 1cm under opaque water). The time taken to locate a hidden platform, the escape latency, was recorded. Mice that did not locate

the platform within 60 seconds were gently guided to the platform and allowed to remain on it for 3 seconds before being removed. Each mouse received 4 trials per session with 10 minutes between each trial. Two sessions separated by 1 hour were conducted per day. After seven sessions, mice underwent a probe trial in which the hidden platform was removed. Time spent in each quadrant of the water bath was recorded. In addition, the cumulative distance the mice swam in search of the hidden platform was recorded.

Conditioned fear testing was used to assess fear-associated memory formation. On the first day of conditioned fear training, mice were placed in a novel fear conditioning chamber for 2 minutes and allowed to explore while baseline freezing time was recorded. The mice were then exposed to a 30 second tone (80dB), which was immediately followed by a 0.4mA foot shock for 2 seconds. Two minutes later, the tone-shock pairing was delivered again. Ten seconds later, the mice were removed from the chamber and returned to their home cages. The following day, mice were placed into the same fear conditioning chamber and their mobility was recorded for 3 minutes in order to assess freezing behavior (cessation of all movement except for respiration). No tones or shocks were delivered. One hour later mice were placed in a novel chamber and the same conditioned fear tone was delivered 3 minutes later. Freezing behavior was recorded during the three minutes prior to exposure to the fear tone and for the three minute period following exposure to the tone.

### Statistical analysis

Simultaneous comparison of neuron death among the genotypes and conditions tested was performed by two-tailed Dunnett's test using the statistical software R (version 2.11.1). Gene expression was quantified using the comparative $C_T$ method in QuantStudio 6 and expressed as mean ± 95% confidence interval. A three way repeated measures ANOVA was used to determine potential group differences in the learning acquisition of the water maze test with sessions as the within subject factors and genotype and isoflurane exposure as the between subject factors. A one way ANOVA was performed for the MWM probe trial to assess within group differences between the target quadrant versus the non-target quadrants with the statistical software SPSS (V. 24). Intergroup differences in the cumulative distance swam away from the center of the target platform was assessed by two-way t-test using R.

## Results

### Bax is necessary for neuron death following neonatal exposure to anesthesia

Neuron death associated with ethanol exposure in early life occurs primarily via apoptosis, and is blocked by genetic deletion of *Bax* [10]. We therefore sought to test the hypothesis that neonatal exposure to isoflurane induces cell death through apoptosis. We developed an exposure paradigm that partially titrates isoflurane delivery to a level of 1 MAC over a 6 hour period. Specifically, P7 pups were exposed to 4% isoflurane for 15 minutes to induce a general anesthetic state, then isoflurane was reduced to 2% for 15 minutes, then further reduced by 0.2% increments every 30 minutes until a final concentration of 1% was reached and maintained for the final 120 minutes (Fig 1A). Animals tolerated this exposure well, with a low mortality rate and rapid recovery to baseline activity following completion of the exposure period. Two hours following completion of the exposure, control and isoflurane treated animals were indistinguishable based on appearance and activity level. At this point, animals were euthanized, and brains were assessed by immunohistochemistry using cleaved caspase-3 as a marker of apoptotic cell death (Fig 1B). We found that in wild-type animals exposed to isoflurane

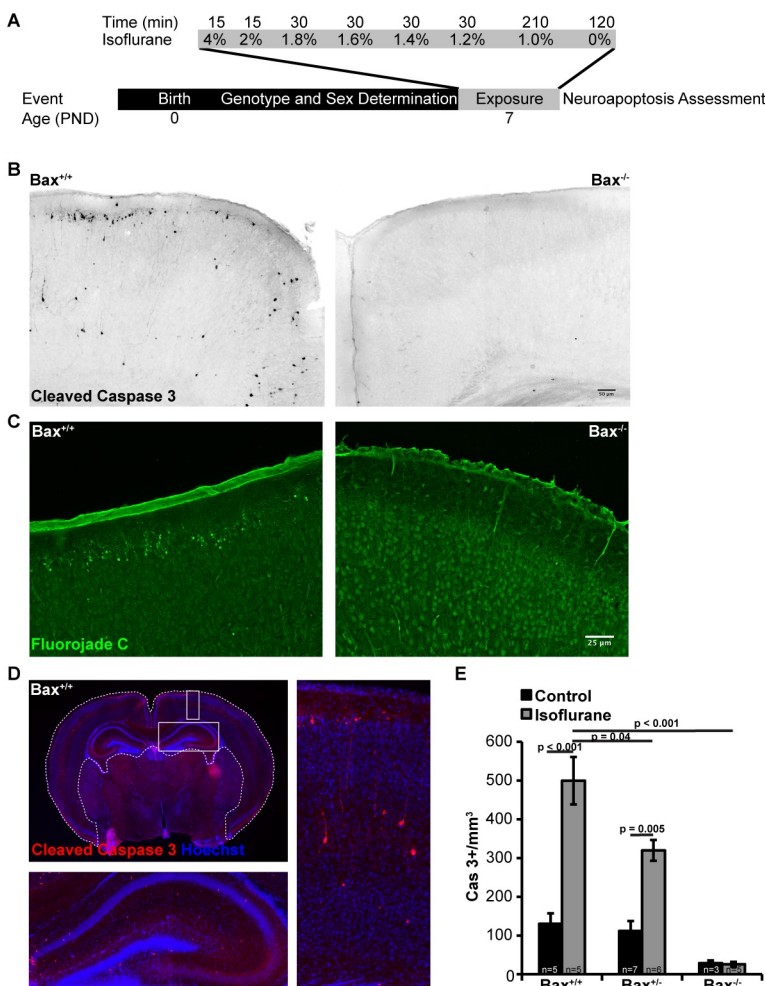

**Fig 1. Neuron death occurs in neonatal animals exposed to isoflurane.** A) diagram of the experimental isoflurane exposure paradigm. On PND 7 mice were exposed to isoflurane which was partially titrated to a level of 1 MAC over the duration of the exposure. The time (in minutes) mice were exposed to each concentration of isoflurane is shown along the top. B) Coronal sections approximate to Paxinos plate P6 #30 from $Bax^{+/+}$ and $Bax^{-/-}$ animals exposed to isoflurane and probed for cleaved caspase 3. C) Coronal sections similar to (B) but stained with Fluorojade C reveals dye accumulation in many neurons in the exposed $Bax^{+/+}$ animals and few in $Bax^{-/-}$ animals. D) Representative brain section labeled for cleaved caspase 3 with the region of interest (dashed line) wherein the number of cleaved caspase 3 positive neuron profiles was counted. Insets showing higher magnification of the hippocampus and cortex are outlined with a solid line. E) Results of the quantification of cleaved caspase 3 neurons profiled within the region of interest indicated in (D).

treatment, cleaved caspase-3 containing neurons were readily detectable throughout the brain. Consistent with previous descriptions, the distribution of cleaved caspase-3 neurons was higher in the superficial cortex and in layer V, where morphological features of pyramidal neurons were apparent [9, 28]. Assessment of cell death in wild-type animals using an unbiased marker of dead and degenerating neurons, FluoroJade C, revealed a pattern of neuron death similar to that seen with cleaved caspase-3 (Fig 1C). This suggests that the majority of dying neurons were doing so via apoptosis. The number of apoptotic cells was quantified by counting the number of cleaved caspase-3 positive cells with neuronal morphology within a region of interest that included the cortex and hippocampus (Fig 1D). This analysis revealed that isoflurane exposure results in a 5-fold increase in cleaved caspase-3 positive neurons compared to control treated wild-type animals (Fig 1E).

In contrast to wild-type mice, *Bax* deficiency afforded protection from apoptotic cell death following exposure to anesthesia in a gene-dose dependent manner (Fig 1B–1E). *Bax*[+/-] animals display a significant reduction in the number of cleaved caspase-3 positive neurons following exposure to isoflurane, with only a 3-fold increase compared to control treated *Bax*[+/-] animals. Homozygous deletion of *Bax* resulted in a nearly complete elimination of cleaved caspase-3 positive neurons following exposure to isoflurane. Control *Bax*[-/-] animals also displayed a level of cell death that is 5-fold lower than control treated wild-type and *Bax*[+/-] animals, confirming that a low level of Bax-dependent cellular pruning is a ubiquitous developmental process. Taken together, these results show that neuron death following neonatal exposure to isoflurane occurs via Bax-dependent apoptosis.

## Neuroinflammation occurs as a consequence of Bax mediated neuron death

Neuroinflammation has been described following exposure to volatile anesthetic exposure in the neonatal period, and it has been proposed that inflammation itself contributes to cognitive deficits later in life [16]. We therefore tested whether neuroinflammation occurs as a consequence on neuron death, or is an independent process. Similar to a previous report of EtOH associated neuroinflammation in neonatal animals, we found that following exposure to isoflurane, Iba1[+] microglia in the hippocampus (Fig 2A) and cortex (Fig 2B) of wild-type mice undergo a clear change in morphology and distribution [15]. Microglia processes appear to retract around the soma, changing from a ramified morphology to an ameboid morphology consistent with their activation [29]. In contrast, in *Bax*[-/-] animals, in which apoptosis is blocked, microglia retain their ramified morphology following exposure to isoflurane. While Iba1 levels appear to change by immunohistochemistry following exposure to isoflurane in wild-type mice, this likely reflects the compaction of microglial morphology, not an increase in microglial number. Consistent with this, the relative amount of Iba1 protein does not change following exposure to isoflurane (Fig 2C).

Microglial activation was further assessed by quantifying the expression of the microglia-derived proinflammatory cytokines TNFα and IL-1β, as well as CR3/MAC-1 β2-integrin subunit (Itgβ2) and purinergic receptor (P2Y12), which are up- and down-regulated, respectively, during microglia activation. As expected, following exposure to isoflurane, the expression of TNFα, IL-1β, and Itgβ2 are all increased, and P2Y12 expression is decreased in wild-type mice (Fig 2D). In contrast, the expression profile of microglia-activation associated genes is attenuated in *Bax*[-/-] animals following isoflurane exposure (Fig 2D). These results demonstrate that microglial activation occurs downstream of Bax-mediated neuron apoptosis following neonatal exposure to anesthesia.

## GABAergic neurons are vulnerable to isoflurane-induced death

GABAergic neurons represent ~15% of the total cortical neuron population in adult rodents. The final population of GABAergic neurons is regulated by a wave of developmentally-regulated, Bax-dependent apoptosis that overlaps with the period of vulnerability to volatile anesthetic toxicity during the early postnatal period [14, 30]. Previous studies have suggested that cortical GABAergic interneurons may be overrepresented among dying cells following exposure to anesthesia [9, 31]. This raises the possibility that an increase in GABAergic neuron death following exposure to anesthesia occurs due to an amplification of the normal wave of developmental apoptosis.

To test this, we first quantified the proportion of GABAergic neurons within the cleaved caspase-3 positive population following exposure to isoflurane. To identify GABAergic neurons, we used a genetic approach by generating a conditional knockout/reporter line,

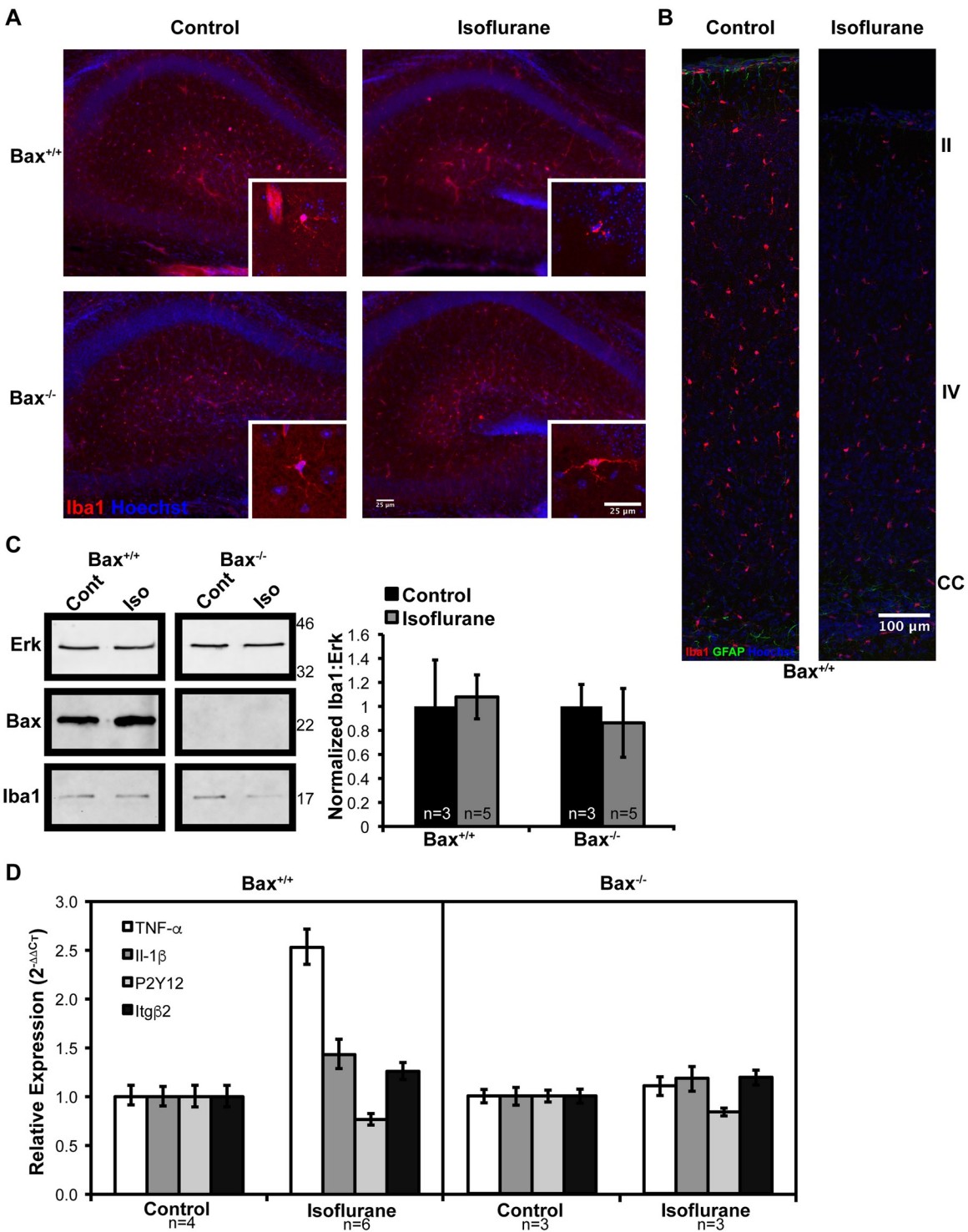

**Fig 2. Neuroinflammation following exposure to isoflurane in the neonatal period is a consequence of neuron death.** A) Low magnification images of hippocampal microglia and high magnification images taken in the molecular layer in the CA1 region (inset). B) Microglia imaged throughout the cortex. Approximate position of cortical layers II, IV and the corpus callosum (CC) are shown. C) Western blot analysis of Iba1 content in cortical lysate and quantification by band densitometry. D) RT-PCR analysis of the relative expression of microglia-derived proinflammatory cytokines and microglia activation genes, mean ± 95% confidence intervals are shown.

$Gad2\text{-}IRES^{Cre}$;$Bax^{Flox}$;$R26^{TdTom/TdTom}$. At PND7, the relative proportion of GABAergic interneurons is ~16–17% of the cortical neuron population and is not affected by conditional deletion of *Bax* from interneurons (Fig 3A). Isoflurane exposure induced apoptosis in both GABAergic and non-GABAergic neurons based on the presence of cleaved caspase-3 and TdTomato co-staining (Fig 3B). Quantification of cell death revealed that in heterozygous $Gad2\text{-}IRES^{Cre}$;$Bax^{Flox/+}$;$R26^{TdTom/TdTom}$ mice, GABAergic interneurons comprise ~30% of the cleaved caspase 3 population (Fig 3C). Therefore, GABAergic neurons are overrepresented by approximately twofold in the population of neurons undergoing apoptosis following exposure to anesthesia. Genetic deletion of *Bax* from interneurons ($Gad2\text{-}IRES^{Cre}$; $Bax^{Flox/Flox}$;$R26^{TdTom/TdTom}$) results in a modest reduction in the proportion of GABAergic neurons undergoing apoptosis to ~15%. Based on our results using conventional $Bax^{-/-}$ mice, we had anticipated complete protection of interneurons from anesthesia-associated death in these mice. However, when we assessed the degree of *Bax* deletion in $Gad2\text{-}IRES^{Cre}$; $Bax^{Flox/Flox}$;$R26^{TdTom/TdTom}$ animals, we found that Bax protein in interneurons was reduced, but not eliminated. This suggests that despite the *Gad2* promoter driving recombination at ~E19 [32], there was still Bax protein present in GABAergic neurons at PND7, possibly due to slow protein turnover (Fig 3D). For this reason, at PND7 the $Gad2\text{-}IRES^{Cre}$;$Bax^{Flox/Flox}$; $R26^{TdTom/TdTom}$ animals are best thought of as a GABAergic interneuron specific *Bax* knockdown, rather than a knock-out model.

## Selective protection of GABAergic neurons alters seizure susceptibility

Prior work suggests that exposure to volatile anesthetics may result in lasting alteration of seizure susceptibility. First, volatile anesthetics cause epileptic discharge-like activity when delivered in sub-burst suppression doses [33, 34]. Furthermore, overactivation of microglia during early development is known to enhance epileptogenicity of neurotoxic insults [35]. Finally, disruption of the excitatory:inhibitory (E:I) ratio through loss of GABAergic interneurons correlates with epilepsy severity [36]. Given our observations that isoflurane exposure activates microglia and that GABAergic neurons are overrepresented among cleaved caspase-3 positive neurons, we tested whether the mice exposed to isoflurane as neonates showed altered seizure susceptibility as adults. We also asked whether reducing inhibitory neuron death in $Gad2\text{-}IRES^{Cre/+}$;$Bax^{Flox/Flox}$ mice offered protection from seizure susceptibility. Seizure induction was done by exposure to Bis-(2,2,2-Trifluoroethyl)-Ether (Flurothyl) due to its ease of administration, demonstrated consistency, and rapid recovery from seizure following discontinuation of the exposure [25, 26].

Upon exposure to Flurothyl, latency to the onset of the first myoclonic jerk was not statistically different between control and isoflurane exposed groups in either control $Gad2\text{-}IRES^{Cre/+}$;$Bax^{Flox/+}$ or $Gad2\text{-}IRES^{Cre/+}$;$Bax^{Flox/Flox}$ (Fig 4A). Similar to the results seen with latency to the first myoclonic jerk, control $Gad2\text{-}IRES^{Cre/+}$;$Bax^{Flox/+}$ mice did not display any increased sensitivity to tonic-clonic seizure (TCS) induction associated with early life exposure to isoflurane. Surprisingly, isoflurane exposed $Gad2\text{-}IRES^{Cre/+}$;$Bax^{Flox/Flox}$ mice demonstrated consistent protection from TCS induction (Fig 4B). Taken together, these results suggest that the increased neuron apoptosis and neuroinflammation observed following early life exposure to isoflurane does not alter susceptibility to seizures later in life.

## Assessment of learning and memory following isoflurane exposure in early life

We found that exposure to isoflurane early in life induced neuron death and neuroinflammation, which could be blocked by constitutive deletion of *Bax*. Therefore, we tested whether

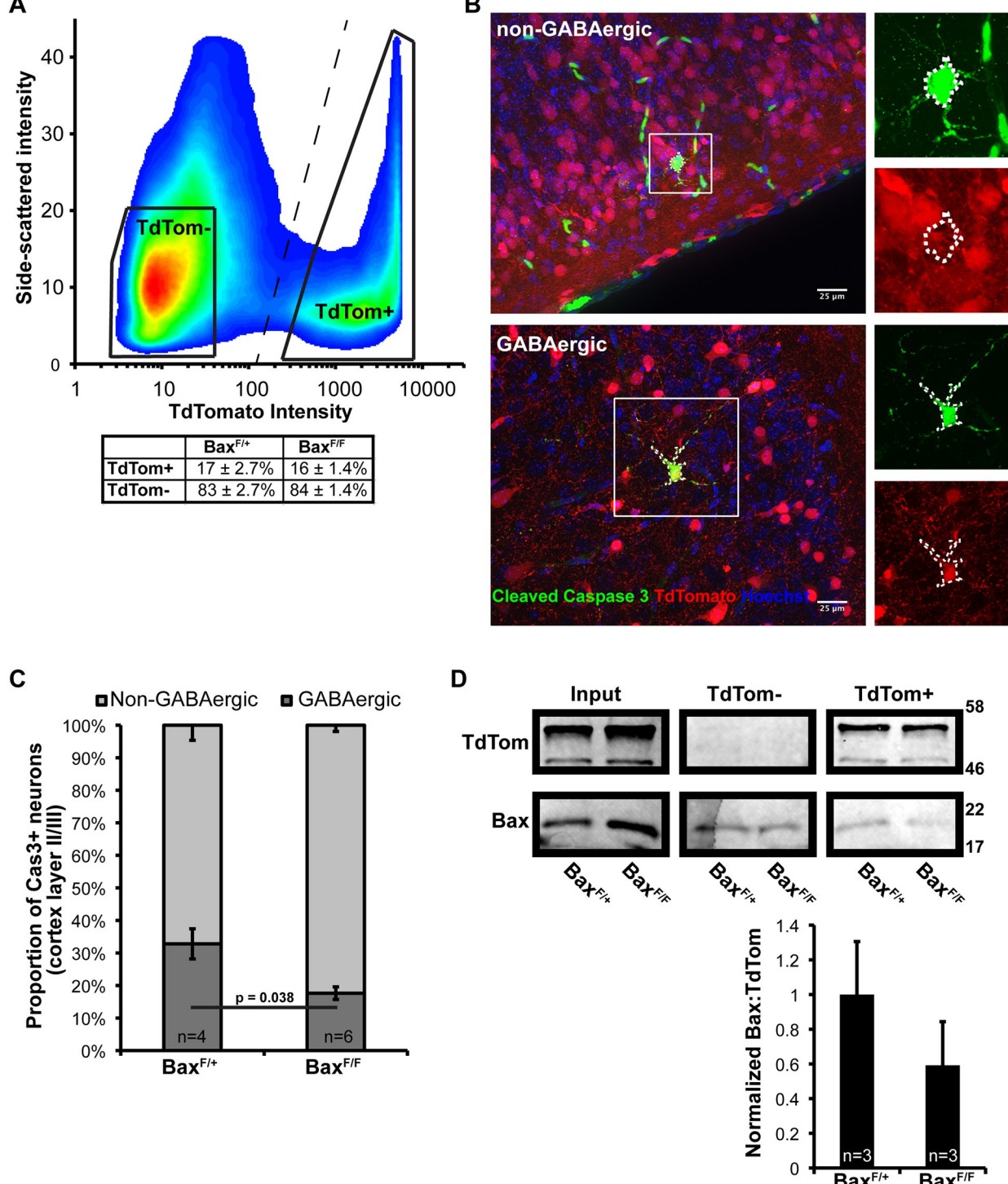

**Fig 3. GABAergic interneuron population size and vulnerability to isoflurane exposure at PND 7.** A) Cortical neurons from the *Gad2-IRES^Cre*; *Bax^Flox*;*R26^TdTom/TdTom* mice were quantified by FACS based on TdTomato fluorescence and side scatter. Demarcation of the GABAergic and nonGABAergic neuron populations for counting is shown as the dashed line. Solid trapezoids are representative of the collection gates used for sorting the cell populations used for subsequent western blot analysis in (D). Three animals per genotype from three separate litters were used for this analysis. B) Representative images of cleaved caspase 3 positive non-GABAergic (top) and GABAergic (bottom) neurons following exposure to isoflurane. C) Quantification of the relative proportion of non-GABAergic and GABAergic neurons in the total cleaved caspase 3 positive population in cortical layers II/III. D) Western blot analysis of cell suspension input for FACS and the isolated populations following sorting with quantification of the relative Bax protein content in the TdTomato positive population by band densitometry.

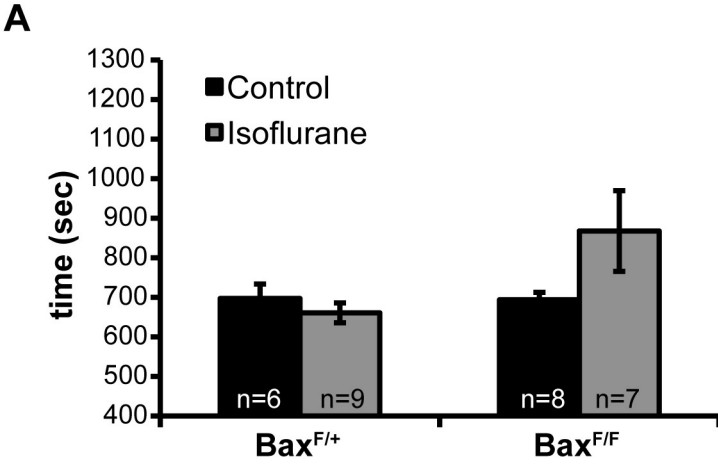

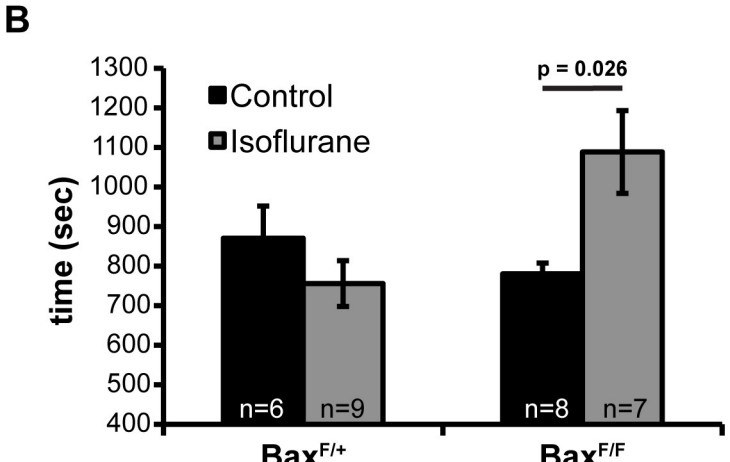

**Fig 4. Seizure susceptibility following early life exposure to isoflurane on PND 7 was assessed using the Gad2-IRES$^{Cre}$;Bax$^{Flox}$;$^{TdTom}$/$_{TdTom}$ line at age PNW 7–8.** A) Latency to the onset of the first myoclonic jerk following the initiation of the exposure to the volatile proconvulsant Bis (2,2,2-Trifluoroethyl) Ether. B) Latency to the onset of TCS with loss of postural control.

neonatal exposure to isoflurane causes cognitive defects later in life, and whether blocking neuron apoptosis in this context provides protection. The Morris Water Maze (MWM) test was used to assess deficits in hippocampal-dependent visual-spatial memory formation following early life exposure to anesthesia [1]. There was no effect on the escape latency during the training period when comparing controls and mice exposed to isoflurane in the $Bax^{+/+}$, $Bax^{+/-}$, and $Bax^{-/-}$ groups (Fig 5A). Unexpectedly, a consistent difference in escape latency between the $Bax^{+/+}$ and $Bax^{-/-}$ animals in the control conditions was observed through the first six training sessions. Differences in swin speed were observed between genotypes on the visible platform sessions (P<0.001). We also analyzed performance using distance moved and found a consistent effect of genotype across the average of all the combined hidden platform sessions (P<0.05). However, all groups showed an effect of session number (P<0.001), indicating that all mice learned the location of the hidden platform. A probe trial was performed 24 hours

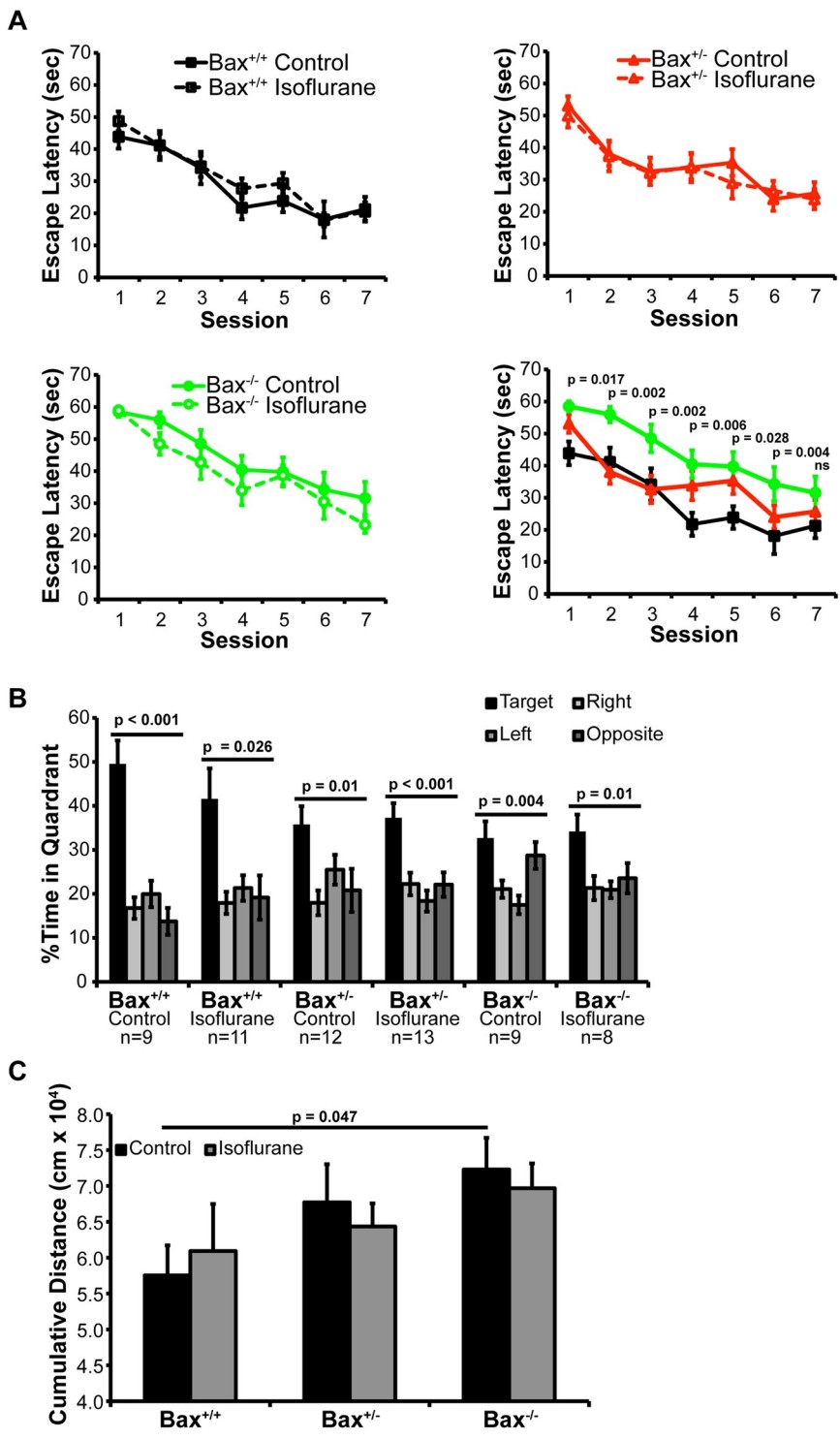

**Fig 5. Hippocampal-dependent visual spatial memory formation was assessed using the Morris-Water Maze.** A) Learning curves on repeated training sessions, p values comparing *Bax*[+/+] and *Bax*[-/-] animals by group sizes are as indicated in (B). Comparison between the Bax[+/+] and Bax[-/-] groups that were significant are indicated. B) Place preference assessed as percent time spent in each quadrant of the maze over the 60 seconds of the probe trial in which the escape platform was removed and the visual cues remained unchanged. C) Cumulative distance the animal swam from where the platform was originally located.

following the training sessions to assess memory retention and recall (Fig 5B). All genetic and treatment conditions displayed place preference for the quadrant formerly containing the hidden platform (the target quadrant). We assessed potential group differences in the memory retention and recall process by evaluating cumulative distance the animal swam from the center point of the hidden platform (Fig 5C). There was no difference between control and isoflurane exposed animals within each genotype group. However, a significant difference between the $Bax^{+/+}$ and $Bax^{-/-}$ groups was observed. Notably, we conducted hidden platform training sessions until all groups of mice showed comparable performance (session 7) prior to assessing their ability to retain the acquired spatial information in the probe test. Thus, differences in swim speed likely do not explain genotype difference on the probe test.

In addition to the MWM, fear conditioning assays have been used extensively to evaluate cognitive deficits in rodents exposed to volatile anesthetics in early life [1]. For this reason we also evaluated this paradigm of learning and memory in $Bax$ knockout animals. We used a standard training regime and tested 24 hours later for contextual and cued fear behavior. During the training period, animal activity was observed for the first two minutes the subjects were in the novel fear conditioning chamber prior to exposure to noxious auditory or electrical stimuli, and no differences in freezing time or mobility among the animals was observed (Fig 6A). One day following training, contextual fear memory was assessed (Fig 6B). We observed no difference between control and isoflurane exposed animals within each genotype group. However, similar to the MWM test, $Bax^{-/-}$ mice from the control group exhibited a weaker memory, characterized by less freezing time. Cued fear memory displayed a similar pattern (Fig 6C). Although the $Bax^{-/-}$ mice did not show a significant increase in freezing upon re-exposure to the tone that was previously associated with the shock, there were no significant interactions between genotype and treatment to suggest that this was independent of its genotype deficit.

Collectively these results show that despite causing increased neuronal apoptosis, early life exposure to isoflurane did not result in deficits in hippocampal-dependent visual spatial memory formation or hippocampal/amygdala-dependent fear conditioning. Interestingly, we did observe defects in $Bax^{-/-}$ mice in the control groups, suggesting that interfering with normal developmental apoptosis resulted in memory-related cognitive deficits.

## Discussion

This study was prompted by the finding that in addition to neuron death, exposure to anesthetic agents in the neonatal period results in disrupted synapse architecture, mitochondrial morphology and axonal pruning [37–39]. We attempted to delineate the contribution of neuron death from other potentially injurious effects associated with exposure to volatile anesthetics in the neonatal period using $Bax$ knockout mice. We found that Bax is essential for neuron death, as $Bax^{-/-}$ mice displayed no evidence of apoptotic or dead/degenerating neurons following exposure to isoflurane at PND7 (Fig 1).

We also evaluated the acute neuroinflammatory response following isoflurane exposure. We reasoned that neuroinflammation may occur as an injurious process independent of neuron death. Neuroinflammation has been described in adult models of anesthesia exposure, but has not been assessed in neonatal models [40]. We therefore investigated whether neuroinflammation occurs independent of neuron death using $Bax$ knockout mice. We found that in the absence of Bax-mediated neuron apoptosis, microglia-mediated neuroinflammation is attenuated (Fig 2). This suggests that during the neonatal period, volatile anesthetics are not a strong proinflammatory stimuli, but rather neuron apoptosis is the primary stimuli which

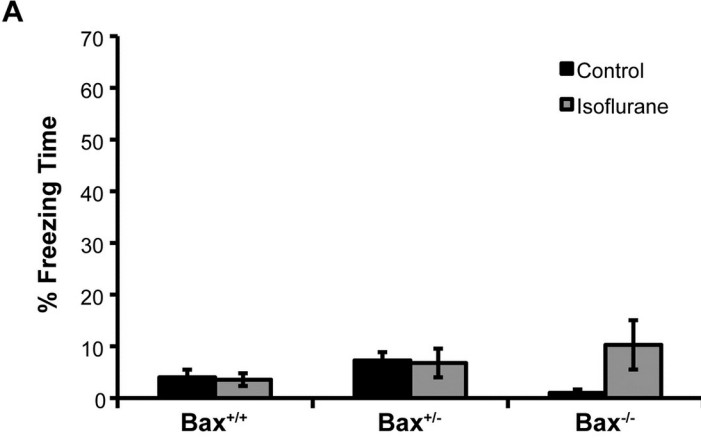

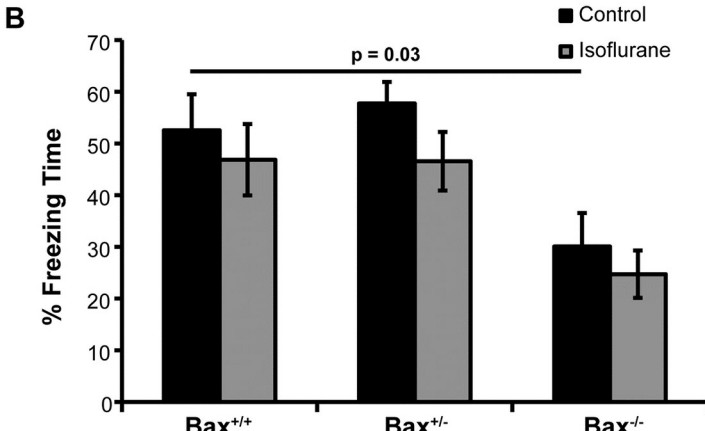

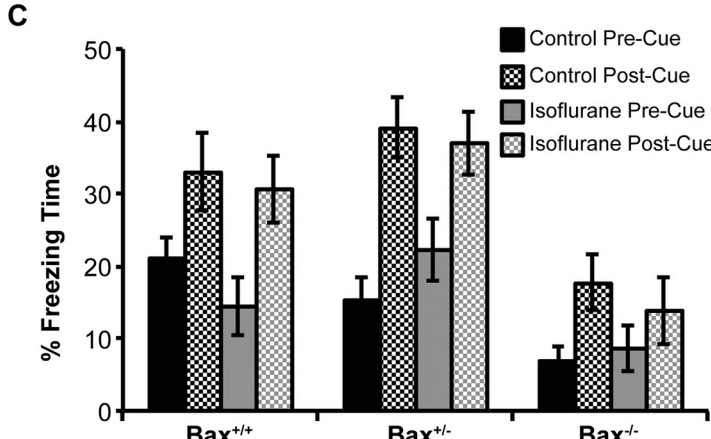

**Fig 6. Hippocampal/Amygdala-dependent fear conditioning assessed with cued and context fear conditioning.** A) Baseline freezing time assessed during the first exposure to the fear conditioning chamber prior to exposure to noxious stimuli. B) 24 hours after training mice were returned to the fear conditioning chamber and freezing time over a 3 minute observation period was assessed. Comparison between the Bax[+/+] and Bax[-/-] groups that were significant are indicated. C) One hour after contextual fear assessment, mice were placed in a novel chamber and exposed to the auditory tone heard during training, freezing time prior to exposure and following the tone was assessed. Sample size for each condition was: Bax[+/+] control n = 9; Bax[+/+] isoflurane n = 9; Bax[+/-] control n = 11; Bax[+/-] n = 13; Bax[-/-] control n = 8; Bax[-/-] isoflurane n = 7.

induces microglial activation. This is similar to the injury and response pattern observed with early life exposure to ethanol [15].

Bax-dependent cellular pruning is a normal part of neurological development, and GABAergic neurons undergo extensive apoptotic pruning during the first two weeks of post-natal life [14]. Cortical GABAergic interneurons are also thought to be particularly vulnerable to volatile anesthetic associated death in the neonatal period [31]. Indeed, we found that GABAergic interneurons are overrepresented within the apoptotic population following exposure to isoflurane (Fig 3). We predicted that excessive interneuron apoptosis could potentially lead to increased susceptibility to seizures. Supporting this hypothesis, epileptogenic-like EEG has been observed as a consequence of early life exposure to volatile anesthetics [41, 42]. However, we found no decrease in seizure threshold associated with early life exposure to volatile anesthetics in control genotype mice (Fig 4). This discrepancy between our study and the previous work may be explained by the specific anesthetic used (sevoflurane vs isoflurane) [42]. Surprisingly, we did find that *Gad2-IRES*$^{Cre/+}$;*Bax*$^{Flox/Flox}$ mice had increased seizure threshold in isoflurane exposed mice, suggesting that selective protection of GABAergic neurons from volatile anesthetic associated death may result in alterations of the global inhibitory-excitatory ratio. Collectively, the findings suggest that volatile anesthetic associated apoptosis may in part represent accelerated pruning of GABAergic interneurons already destined to be eliminated during development.

We attempted to investigate whether the neuronal apoptosis and neuroinflammation following early life exposure to volatile anesthetics had lasting consequences on cognitive function later in life. To our surprise, we found that *Bax* deficiency itself led to deficits in multiple aspects of cognition, independent of volatile anesthetic exposure. Previous studies have described *Bax* knockout mice as having persistently prolonged escape latency in MWM training sessions when visual spatial learning was assessed [43, 44]. The deficits we observed suggest that normal Bax-mediated developmental apoptosis in early life is critical for cognitive development. Cellular pruning, synapse remodeling, and axon pruning are all processes known to be mediated by Bax-dependent caspase activation, and disruption of any or all could give rise to negative effects on cognitive development [14, 45, 46]. We cannot exclude the possibility that the lack of cognitive deficits in wild-type animals exposed to isoflurane treatment in early life is due to specifics of the study design. However, if the neuron apoptosis and neuroinflammation attributable to volatile anesthetic exposure in early life does contribute to cognitive defects later in life, it is relatively mild when compared to the effect that blocking normal developmental caspase activation with *Bax* deficiency has on cognitive development.

In conclusion we have demonstrated that Bax is necessary for neuron death associated with early life exposure to volatile anesthetics. The neuroinflammation seen following volatile anesthetic exposure in the neonatal period likely arises as a secondary consequence of Bax-mediated neuron death rather than being an independent response to volatile anesthetic exposure. And finally, GABAergic interneurons are overrepresented among the dead neurons, suggesting they are more susceptible to the pro-apoptotic effects of volatile anesthesia. Due to the cognitive deficits attributable to *Bax* deficiency alone, we were unable to conclusively determine whether blocking apoptosis and neuroinflammation following volatile anesthetic exposure provided any benefit with respect to cognitive function. Establishing a transient protected state through the use of Bax- or caspase-specific small molecule inhibitors may prove to be a viable alternative approach for investigating lasting consequences of early life exposure to volatile anesthetics. Further investigation along these lines will be necessary to delineate the contribution of neuron death from other disruptions in neuronal function to the injury arising from early life exposure to volatile anesthetics.

## Supporting information

**S1 Raw images.**
(TIF)

## Acknowledgments

The authors thank Kylee Rosette, B.S., for technical assistance with mouse colony maintenance and anesthesia assays and Mike Jacobson, B.S., for technical assistance with behavioral assays.

## Author Contributions

**Conceptualization:** Andrew M. Slupe, Kevin M. Wright.

**Data curation:** Andrew M. Slupe.

**Formal analysis:** Andrew M. Slupe, Laura Villasana, Kevin M. Wright.

**Funding acquisition:** Andrew M. Slupe, Kevin M. Wright.

**Investigation:** Andrew M. Slupe, Laura Villasana.

**Methodology:** Andrew M. Slupe, Laura Villasana.

**Project administration:** Kevin M. Wright.

**Supervision:** Kevin M. Wright.

**Visualization:** Andrew M. Slupe.

**Writing – original draft:** Andrew M. Slupe, Kevin M. Wright.

**Writing – review & editing:** Andrew M. Slupe, Laura Villasana, Kevin M. Wright.

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
