## [Decision Letter · Decision Letter 0]

23 Oct 2020

PONE-D-20-27023

BAX is necessary for neuronal death following exposure to isoflurane during the neonatal period

PLOS ONE

Dear Dr. Wright,

Thank you for submitting your manuscript to PLOS ONE. After careful consideration, we feel that it has merit but does not fully meet PLOS ONE’s publication criteria as it currently stands. Therefore, we invite you to submit a revised version of the manuscript that addresses the points raised during the review process.

We look forward to receiving your revised manuscript.

Kind regards,

Alexandra Kavushansky, PhD

Academic Editor

PLOS ONE

Additional Editor Comments:

In addition to the corrections requested by the Reviewer, i'd like to ask to state which statistical test was performed to verify the normality of the data distribution and which version of the SPSS software was used.

Journal Requirements:

Reviewers' comments:

Reviewer's Responses to Questions

**Comments to the Author**

1. Is the manuscript technically sound, and do the data support the conclusions?

Reviewer #1: Partly

2. Has the statistical analysis been performed appropriately and rigorously? 

Reviewer #1: Yes

3. Have the authors made all data underlying the findings in their manuscript fully available?

Reviewer #1: Yes

4. Is the manuscript presented in an intelligible fashion and written in standard English?

Reviewer #1: Yes

5. Review Comments to the Author

Reviewer #1: Andrew et al. described neonatal exposure to isoflurane caused neuronal death via BAX induced apoptotic pathway and the most apoptotic neurons were GABAergic interneurons. This apoptosis have little effect on seizure threshold or cognitive function later in life. I have several concerns about this study.

Title:

The authors should modify the title to clearly represent their data. Only Fig1 & 2 were relative to the present title. As BAX is a well-known candidate in apoptotic pathway, the interesting part of this study was that BAX most affect GABAergic interneurons and it did not affect later life behaviors.

Abstract:

1. Non-human primate is no need to be mentioned.

2. The conclusion should be more concrete and accurate.

Methods:

1. Total cell number collected per sample in FACS should be mentioned.

2. Chamber size for seizure assay is missing.

3. Why the mice used for seizure assay were 7-8w while mice used for MWM were 10-11w? Are they the same group? if not, the authors should describe it clearly.

4. The age of the mice used for conditioned fear test is missing.

Results:

1. Fig1A should be presented in a clearer way. The recent seems the authors had 8 groups for different test of iso.

2. Overall view of Fig1B & C must be presented.

3. Which area is used for analysis for that in Fig1E?

4. Only immunofluorescence of Caspase3 to draw the conclusion that neuronal death occurs via Bax-dependent apoptosis is weak. More evidence is needed, e.g, Flow cytometry of Annexin V/PI to detect apoptotic neuron.

5. The authors draw the conclusion that microglial activation occurs downstream of Bax-mediated neuron apoptosis seems no sufficient evidence. In Fig2B the microglia looks even deactivated after iso in Bax+/+ and no Bax-/- mice data is presented

6. Fig2C Why use Erk as internal control?

7. P value is missed in Fig2D.

8. As mentioned, total number of cell collected in flow cytometry should be presented in Fig3A and also Flow cytometry of Annexin V/PI to further detect apoptotic neuron.

9. Why the authors only test cortical GABAergic neurons, how about the hippocampus?

10. The internal control is missing in Fig3D.

11. It seems that the baseline of Bax+/+, +/- and -/- in MWM has difference, thus the difference caused by iso should be carefully described.

12. Swim speed is missing.

13. n is missing in Fig6.

Discussion:

The authors should discuss more about why early GABAergice neuron apoptosis had no effect on later life behavior.

6. PLOS authors have the option to publish the peer review history of their article (what does this mean?). If published, this will include your full peer review and any attached files.

Reviewer #1: No

---

## [Author Response · Author response to Decision Letter 0]

18 Dec 2020

In addition to the corrections requested by the Reviewer, i'd like to ask to state which statistical test was performed to verify the normality of the data distribution and which version of the SPSS software was used.

We used Version 24 of SPSS (now indicated in the methods). Data normality was assumed.

Journal Requirements:

We have removed the reference to this data, as the findings were not significant and the data was not central to the study.

Reviewers' comments:

Reviewer's Responses to Questions

Comments to the Author  1. Is the manuscript technically sound, and do the data support the conclusions?  The manuscript must describe a technically sound piece of scientific research with data that supports the conclusions. Experiments must have been conducted rigorously, with appropriate controls, replication, and sample sizes. The conclusions must be drawn appropriately based on the data presented.

Reviewer #1: Partly

2. Has the statistical analysis been performed appropriately and rigorously?

Reviewer #1: Yes

3. Have the authors made all data underlying the findings in their manuscript fully available?  The PLOS Data policy requires authors to make all data underlying the findings described in their manuscript fully available without restriction, with rare exception (please refer to the Data Availability Statement in the manuscript PDF file). The data should be provided as part of the manuscript or its supporting information, or deposited to a public repository. For example, in addition to summary statistics, the data points behind means, medians and variance measures should be available. If there are restrictions on publicly sharing data—e.g. participant privacy or use of data from a third party—those must be specified.

Reviewer #1: Yes

4. Is the manuscript presented in an intelligible fashion and written in standard English?  PLOS ONE does not copyedit accepted manuscripts, so the language in submitted articles must be clear, correct, and unambiguous. Any typographical or grammatical errors should be corrected at revision, so please note any specific errors here.

Reviewer #1: Yes

5. Review Comments to the Author  Please use the space provided to explain your answers to the questions above. You may also include additional comments for the author, including concerns about dual publication, research ethics, or publication ethics. (Please upload your review as an attachment if it exceeds 20,000 characters)

Reviewer #1: Andrew et al. described neonatal exposure to isoflurane caused neuronal death via BAX induced apoptotic pathway and the most apoptotic neurons were GABAergic interneurons. This apoptosis have little effect on seizure threshold or cognitive function later in life. I have several concerns about this study.

We thank the reviewer for their thoughtful comments and suggestions on the study. We have attempted to address each of the specific critiques on a point-by-point basis below.

Title:

The authors should modify the title to clearly represent their data. Only Fig1 & 2 were relative to the present title. As BAX is a well-known candidate in apoptotic pathway, the interesting part of this study was that BAX most affect GABAergic interneurons and it did not affect later life behaviors.

Title has been changed to "GABAergic neurons are susceptible to BAX-dependent apoptosis following isoflurane exposure in the neonatal period" to address this concern. We feel that this revised title now represents the strongest conclusions from the study.

Abstract:

1. Non-human primate is no need to be mentioned.

'Rodent and Non-human primate' have been removed from the abstract.

2. The conclusion should be more concrete and accurate.

We have re-written the abstract to better describe the outcomes of the behavioral studies and believe it now more accurately reflects the main findings of the study.

Methods:

1. Total cell number collected per sample in FACS should be mentioned.

The range of total cell numbers collected (450K - 1000K) is now included in the text of the methods section (page 11, red text),

2. Chamber size for seizure assay is missing.

Chamber size is now included in the methods section (page 12, red text). We can also provide a photograph of the chamber in the supplemental data if necessary.

3. Why the mice used for seizure assay were 7-8w while mice used for MWM were 10-11w? Are they the same group? if not, the authors should describe it clearly.

We apologize for this confusion. The mice used for the seizure assays are a completely different cohort from the mice used for behavioral assays. This is now explicitly stated on page 12. The respective ages for these cohorts were chosen to be consistent with the prior publications cited in the methods section that were used to guide the experiments (indicated in red text, page 12-13). 

4. The age of the mice used for conditioned fear test is missing.

Conditioned fear testing was performed at PNW 11. This is now indicated in red text at the end of the first paragraph of the 'Behavioral Assays' section in the Methods (page 13).

Results:

1. Fig1A should be presented in a clearer way. The recent seems 

the authors had 8 groups for different test of iso.

We apologize for this lack of clarity. A more detailed description of the exposure paradigm is included in the methods and results section (red text, page 6, page 15) to avoid causing confusion. 

2. Overall view of Fig1B & C must be presented.

Unfortunately, we do not have larger widefield images showing the entire brain. The imaging for these experiments was performed at 10x and 20x in order to clearly identify Cleaved Caspase-3 and Fluorojade C positive neurons based on their morphology, respectively. 

3. Which area is used for analysis for that in Fig1E?

The region of interest indicated by the dashed lines in Fig1D used for the quantification in Fig1E. This is now described more clearly in the figure legend.

4. Only immunofluorescence of Caspase3 to draw the conclusion that neuronal death occurs via Bax-dependent apoptosis is weak. More evidence is needed, e.g, Flow cytometry of Annexin V/PI to detect apoptotic neuron.

We respectfully disagree with the reviewer, and believe that the data presented using both Cleaved Caspase 3 and Fluorojade C is sufficient to conclude that apoptotic cell death is occurring. Furthermore, the fact that this is eliminated in Bax-/- mice (Figure 1B-E) provides strong genetic evidence for the conclusion that isoflurane induced neuronal death occurs via Bax-dependent apoptosis.

5. The authors draw the conclusion that microglial activation occurs downstream of Bax-mediated neuron apoptosis seems no sufficient evidence. In Fig2B the microglia looks even deactivated after iso in Bax+/+ and no Bax-/- mice data is presented

The data presented in Fig 2 is consistent with prior reports of ethanol associated microglia activation (see Ahlers, 2014). In this paradigm, microglial activation is assessed by a change in microglial morphology from a ramified to ameboid shape, not by the levels of Iba1 fluorescence. This microglial “compaction” likely accounts for the perceived differences in Iba1 fluorescence in Fig 2B. This is now more directly referenced and clarified in the relevant results section (page 17-18). Furthermore, the conclusion that microglial activation is downstream of neuronal apoptosis is supported by the increased expression of microglial-derived proinflammatory cytokines, which is blocked in Bax-/- mice (Figure 2D).

6. Fig2C Why use Erk as internal control?

Erk was chosen as an internal control as its expression has previously been demonstrated to be insensitive to fetal alcohol exposure (Samudio-Ruiz, et al, Journal of Neurochemistry, 2009). The injury mechanism of fetal alcohol exposure is similar to that of volatile anesthetics. We have added a sentence indicating this in the methods section (red text, page 9).

7. P value is missed in Fig2D.

Due to the method of data analysis, exact p values are not generated. Rather 95% confidence intervals are shown. Non-overlapping 95% CIs suggest significant differences in expression.

8. As mentioned, total number of cell collected in flow cytometry should be presented in Fig3A and also Flow cytometry of Annexin V/PI to further detect apoptotic neuron.

Total number of collected cells is now included in the methods section (red text, page 11). Unfortunately Annexin V/PI counts were not included in these experiments, and including this data would require us to completely re-run the experiments with additional animals, which we no longer maintain in our colony.

9. Why the authors only test cortical GABAergic neurons, how about the hippocampus?

Immunofluorescence imaging analysis of GABAergic neurons was limited to the cortex because this is the population analyzed by FACS. Unfortunately the hippocampus was not prepared or analyzed in these experiments, and similar to the comment above, would require us to re-run the experiments with additional animals that we no longer maintain.

10. The internal control is missing in Fig3D.

TdTom was used as the internal control for the purposes of quantification.

11. It seems that the baseline of Bax+/+, +/- and -/- in MWM has difference, thus the difference caused by iso should be carefully described.

We have now added a description of these results (page 23-24, red text).

12. Swim speed is missing.

See answer to point 11.

13. n is missing in Fig6.

Sample size is now included in the figure legend (red text, page 25). We chose to put this in the legend rather than the figure itself to avoid crowding. The same n's are applicable to (A), (B) and (C).

Discussion:

The authors should discuss more about why early GABAergice neuron apoptosis had no effect on later life behavior.

In the revised manuscript we have added the following to the discussion (Page 28, red text): "Collectively, the findings suggest that volatile anesthetic associated apoptosis may in part represent accelerated pruning of GABAergic interneurons already destined to be eliminated during development."

---

## [Decision Letter · Decision Letter 1]

28 Dec 2020

GABAergic neurons are susceptible to BAX-dependent apoptosis following isoflurane exposure in the neonatal period

PONE-D-20-27023R1

Dear Dr. Wright,

We’re pleased to inform you that your manuscript has been judged scientifically suitable for publication and will be formally accepted for publication once it meets all outstanding technical requirements.

Kind regards,

Alexandra Kavushansky, PhD

Academic Editor

PLOS ONE

Additional Editor Comments (optional):

Reviewers' comments:

Reviewer's Responses to Questions

**Comments to the Author**

1. If the authors have adequately addressed your comments raised in a previous round of review and you feel that this manuscript is now acceptable for publication, you may indicate that here to bypass the “Comments to the Author” section, enter your conflict of interest statement in the “Confidential to Editor” section, and submit your "Accept" recommendation.

Reviewer #1: All comments have been addressed

2. Is the manuscript technically sound, and do the data support the conclusions?

Reviewer #1: Yes

3. Has the statistical analysis been performed appropriately and rigorously? 

Reviewer #1: Yes

4. Have the authors made all data underlying the findings in their manuscript fully available?

Reviewer #1: Yes

5. Is the manuscript presented in an intelligible fashion and written in standard English?

Reviewer #1: Yes

6. Review Comments to the Author

Reviewer #1: The authors have addressed my concerns. It would be better if authors could observe the hippocampus GABAergic neurons.

7. PLOS authors have the option to publish the peer review history of their article (what does this mean?). If published, this will include your full peer review and any attached files.

Reviewer #1: No

---

## [Editor Report · Acceptance letter]

2 Jan 2021

PONE-D-20-27023R1 

GABAergic neurons are susceptible to BAX-dependent apoptosis following isoflurane exposure in the neonatal period 

Dear Dr. Wright:

I'm pleased to inform you that your manuscript has been deemed suitable for publication in PLOS ONE. Congratulations! Your manuscript is now with our production department. 

Kind regards, 

on behalf of

Dr. Alexandra Kavushansky 

Academic Editor

PLOS ONE